# *Pleurotus Ostreatus* Ameliorates Obesity by Modulating the Gut Microbiota in Obese Mice Induced by High-Fat Diet

**DOI:** 10.3390/nu14091868

**Published:** 2022-04-29

**Authors:** Yanzhou Hu, Jia Xu, Yao Sheng, Junyu Liu, Haoyu Li, Mingzhang Guo, Wentao Xu, Yunbo Luo, Kunlun Huang, Xiaoyun He

**Affiliations:** 1Key Laboratory of Precision Nutrition and Food Quality, Key Laboratory of Functional Dairy, Ministry of Education, College of Food Science and Nutritional Engineering, China Agricultural University, Beijing 100083, China; huyz369@163.com (Y.H.); xujia1995012@126.com (J.X.); shengyao314@163.com (Y.S.); liujunyu21@mails.tsinghua.edu.cn (J.L.); kglihaoyu@sina.com (H.L.); guomingzhang126@126.com (M.G.); lyb@cau.edu.cn (Y.L.); 2Key Laboratory of Precision Nutrition and Food Quality, Department of Nutrition and Health, China Agricultural University, Beijing 100083, China; xuwentao@cau.edu.cn; 3Key Laboratory of Safety Assessment of Genetically Modified Organism (Food Safety), The Ministry of Agriculture and Rural Affairs of the P.R. China, Beijing 100083, China

**Keywords:** *Pleurotus Ostreatus*, obesity, gut microbiota, 16S rRNA gene, PICRUSt algorithm

## Abstract

*Pleurotus ostreatus* (PO), a common edible mushroom, contains rich nutritional components with medicinal properties. To explore the effect of PO on ameliorating obesity and modulating the gut microbiota, we administered the mice with a low-fat diet or high-fat diet containing different dosages of PO (mass fraction: 0%, 2.5%, 5% and 10%). The body weight, adipose tissue weight, GTT, ITT, blood lipids, serum biomarkers of liver/kidney function, the gut microbiota and function were measured and analyzed after 6 weeks of PO treatment. The results showed PO prevented obesity, maintained glucose homeostasis and beneficially modulated gut microbiota. PO modified the composition and functions of gut microbiota in obese mice and make them similar to those in lean mice, which contributed to weight loss. PO significantly increased the relative abundance of *Oscillospira*, *Lactobacillus* group and *Bifidobacterium*, while decreased the relative abundance of *Bacteroides* and *Roseburia*. The prediction of gut microbiota function showed PO upregulated lipid metabolism, carbohydrate metabolism, bile acid biosynthesis, while it downregulated adipocytokine signaling pathway and steroid hormone biosynthesis. Correlation analysis further suggested the potential relationship among obesity, gut microbiota and the function of gut microbiota. In conclusion, all the results indicated that PO ameliorated obesity at least partly by modulating the gut microbiota.

## 1. Introduction

Nowadays, obesity has become a global pandemic and is still getting worse [1] because of high caloric intake, unhealthy lifestyle and environmental factors [2]. The World Health Organization (WHO) defined obesity as having a body mass index (BMI) of 30 kg/m^2^ or more [3]. The worldwide population of obese people increased from 111 million (girls 5 million, boys 6 million, adult women 69 million and adult men 31 million) in 1975 to 795 million (girls 50 million, boys 74 million, adult women 390 million and adult men 281 million) in 2016 and there were an additional 1513 million people in the overweight range in 2016 [4]. For obese people, the adipose tissue shows a dysfunctional phenomenon, which affects the hormones and cytokines produced by adipocytes that will expand to the central nervous system, liver, muscle, bone and other organs and tissues [5,6]. Therefore, obesity can induce a series of metabolic diseases [7], including type 2 diabetes, hepatobiliary disease, cardiovascular diseases, osteoarthritis [8], and increase the risk of mortality [9,10]. Actually, humans realized the harm of obesity long time ago. Hippocrates cautioned that sudden death was more likely to occur in obese people than in lean people 2500 years ago [1]. In 2019, *The State of Food Security and Nutrition in the World* report stated that obesity kills about 4 million people every year. So obesity has become a great threat to human health, which is urgent to be resolved worldwide. At present, the main way to lose weight is to reduce energy intake and increase physical activity, but very few people can make it because of the abundant food and little spare time. Therefore, searching for more effective and easier strategies to ameliorate obesity becomes necessary.

More and more studies have confirmed a close relationship between gut microbiota and obesity [11,12,13]. Studies have shown significant differences in intestinal microbiota between obese and lean individuals [14,15]. Gut microbiota transplantation experiments have proved that gut microbiota can change the state of obesity or thinness [16]. It was originally reported that the ratio of Bacillota/Bacteroidota was higher in obese individuals, which would decrease after weight loss, but some subsequent studies did not agree with it, making this view challenged [17,18]. Further, many intestinal bacteria at the genus level have been found to play an important role in preventing obesity, including *Lactobacillus* group, *Oscillospira*, *Bifidobacterium*, *Saccharomyces*, *Streptococcus* and *Enterococcus* [19,20,21]. Gut microbiota can regulate food intake by the gut microbiota-brain axis [22] and regulate energy metabolism, glucose homeostasis, insulin resistance and the immune system through microbial products, especially short chain fatty acids (SCFAs) [17,23] and membrane proteins [24,25]. Moreover, gut microbiota can also promote the thermogenesis of brown adipose tissue (BAT) and browning of white adipose tissue (WAT) through SCFAs [26] and bile acids [27,28]. Gut microbiota can be regulated by diet, which makes preventing obesity by modulating the composition of gut microbiota safe and effective.

*Pleurotus ostreatus* (PO) is a kind of common gray edible mushroom with rich nutritional components and medicinal properties [29,30] and it contains a series of functional components, such as phenolic compounds, flavonoids, saponins, alkaloids, tannins [31]. It has been reported that some mushrooms are beneficial to ameliorate obesity, diabetes, hypertension, atherosclerosis, anemia and constipation [29,30,32,33]. Studies have reported that a mixture of various mushrooms (*Flammulina velutipes*, *Hypsizygus marmoreus*, *Lentinusedodes*, *Grifola frondose*, *Pleurotus eryngii*) suppressed visceral fat accumulation [34]. Our previous research found that *Pleurotus citrinopileatus* has an anti-obesity effect [33]. PO has been confirmed to have functions of lowering blood lipid and ameliorating atherosclerosis, which was achieved by increasing total lipid and cholesterol excretion in feces [35]. Howere, the function of PO in ameliorating obesity has not been reported. PO supplementation can regulate gut microbiota and increase microbial diversity and synthesis of SCFAs in piglets [36]. Besides, PO also has the functions of cancer suppression [37], antioxidation [38], enhancing the immune and protecting liver, kidneys, brain and lungs [39].

In this research, we explored the effect of PO on ameliorating obesity induced by a high-fat diet and modulating the composition of gut microbiota. The relationship between the alteration of gut microbiota and weight loss was particularly analyzed. This paper provides a preliminary research basis to curb obesity using PO.

## 2. Materials and Methods

### 2.1. Material

The PO was purchased from the Beijing Jingnan mushroom planting base (Fangshan, Beijing, China), then it was dried (45 °C, 48 h) and ground to powder. The PO powder was added into the high-fat diet pellets. The composition of PO powder was measured according to GB 5009.3-2016, GB 5009.4-2016, GB 5009.5-2016, GB 5009.6-2016, GB 5009.10-2016, NY/T 1676-2008 and the results are shown in Appendix A.

### 2.2. Animal Experimental Procedures

All the animal experimental procedures were conducted following the Guide for the Care and Use of Laboratory Animals published by the US National Institutes of Health and approved by Animal Ethics Committee of China Agricultural University, Beijing (the approval ID of this study is KY1700014).

Four-week-old C57BL/6J male mice (*n* = 40) free from specific pathogens were purchased from Vital River Laboratories (Beijing, China) and housed under 22 ± 2 °C temperature and 55% ± 10% humidity with 12 h light/12 h dark cycle in a Specific Pathogen Free (SPF) animal room. After a week of acclimatization with free access to a normal commercial basic diet and sterile water, one group of mice were fed with low-fat diet (LFD, 10% calories from fat), while the rest were fed with high-fat diet (HFD, 60% calories from fat) (Appendix A). The mice fed with LFD were marked as positive control group (LFD, *n* = 8). After 6 weeks, the mice fed with HFD were randomly divided into 4 groups based on body weight using a stratified randomization procedure: The negative control group (HFD, *n* = 8) was fed with HFD; Three treated groups were fed with HFD added with PO powder at different mass fractions: 2.5% (HFD + POL, *n* = 8), 5% (HFD + POM, *n* = 8) and 10% (HFD + POH, *n* = 8). In order to fully explore the effects of different dosages of PO, the above three dosages were determined according to relevant studies and our previous studies [33,34]. All the mice had free access to sterile water. The experiment continued for another 6 weeks. The body weight and food intake were measured weekly. In the last week, all mice were sacrificed by cervical dislocation after anesthesia utilizing chloral hydrate.

### 2.3. Blood Biochemistry

The blood was collected from the infraorbital venous plexus of mice and the serum was separated by centrifugation at 3000 rpm for 15 min. The blood biochemical parameters including triglyceride (TG), total cholesterol (CHO), high density lipoprotein (HDL), low density lipoprotein (LDL), blood glucose, alanine aminotransferase (ALT), aspartate aminotransferase (AST), alkaline phosphatase (ALP), total protein (TP), albumin (ALB), uric acid (UA), blood urea nitrogen (BUN) and creatinine (CRE) were determined using a RA-1000 autoanalyzer (Technicon, Concord, NC, USA).

### 2.4. Glucose Tolerance Test (GTT) and Insulin Tolerance Test (ITT)

Obesity can lead to an imbalance of glucose homeostasis. To explore the effect of PO on glucose homeostasis, GTT and ITT were measured.

In the last week of the experiment, GTT was performed after a 16-h fasting. The mice were intraperitoneally injected with glucose (1.5 g/kg BW, sigma). The blood glucose concentration was measured at 0, 15, 30, 60, 90 and 120 min after injection via tail vein blood using a blood glucose meter (Accu-Chek, Basel, Switzerland).

In the last week of the experiment, ITT was performed after a 6-hour fasting. The mice were intraperitoneally injected with insulin (1 IU/kg BW, Novo Nordisk, Copenhagen, Denmark), followed by the measurement of blood glucose concentration at 0, 15, 30, 45 and 60 min after injection.

### 2.5. Gut Microbiota Analysis

The gut microbiota was analyzed at the end of 6 weeks of PO intervention. The fecal samples were collected and sequencing as previously reported [40]. Briefly, the microbial genomic DNA was extracted using a fecal DNA isolation kit (FUDEAN, Beijing, China). The quality and quantification of DNA were assessed with agarose gel electrophoresis. The V3-V4 region of the 16S rRNA was amplified by PCR using universal primers (338 forward: 5′-ACTCCTACGGGAGGCAGCAG-3′; 806 reverse: 5′-GGACTACHVGGGTWTCTAAT-3′) and the libraries were prepared. Sequencing was performed in a HiSeq platform (Illumina, San Diego, CA, USA) at Novogene Bioinformatics Institute (Beijing, China). The raw reads were spliced and filtered to obtain the clean reads. The operational taxonomic units (OTUs) clustering was performed by Uparse software (Uparse v7.0.1001, Robert C. Edgar, Tiburon, CA, USA) [41] and sequences with ≥97% similarity were clustered to the same OTUs. Taxonomic annotation was conducted using the ribosomal database project (RDP) classifier (Version 2.2, Qiong Wang, East Lansing, MI, USA). Based on the sample with minimum bacterial abundance, the data of each sample were normalized to obtain the relative abundances which were used to further analysis.

Alpha diversity, beta diversity, principal component analysis (PCA) and non-metric multidimensional scaling (NMDS) were performed by Past3 software. Alpha and beta diversity were used to determine the species richness and evenness of gut microbiota. The NMDS analysis was used to explore the differences in microbial community structure among different groups. Heat map was made by Heml software to show the distribution of microbial species in different samples. LEfSe analysis was conducted online [42], which was used to find the biomarkers with statistical differences among different groups. The correlation coefficient between microbial relative abundance and the body weight of mice was evaluated using Spearman’s correlation analysis (SPSS Statistic 17.0, IBM, Armonk, NY, USA). Based on 16S rRNA data and the Greengene database, bacterial gene functions were predicted with Phylogenetic Investigation of Communities by Reconstruction of Unobserved States (PICRUSt) [43].

### 2.6. Statistical Analysis

All data were presented as means ± SEM and calculated by GraphPad Prism 5 Software (GraphPad, San Diego, CA, USA). Comparisons between two groups were performed using a two-tailed Student’s *t*-test, while comparisons among more groups were performed using one-way analysis of variance (ANOVA) with Tukey’s multiple comparison test. The significant difference level was *p* < 0.05.

## 3. Results

### 3.1. PO Prevents HFD-Induced Obesity in Mice

After modeling for 6 weeks, the body weight (31.0 ± 0.3) of mice fed with HFD was significantly (*p* < 0.01) higher than that (26.2 ± 0.6) in the LFD group. Then the mice were administered with PO at different dosages. After another 6 weeks, PO lowered the body weight in HFD + POL, HFD + POM and HFD + POH (Figure 1A) without significantly affecting the food intake (Appendix A). The treatment of PO at different dosages all significantly reduced the body weight gain, and the middle- and high-dose groups had lower body weight than the low dose group. The body weight in HFD + POM and HFD + POH was similar to that in LFD group after PO treatment for 5 weeks. Moreover, administration of PO significantly decreased the weight of adipose tissue (Figure 1B). The ratios of epididymal white adipose tissue (EP) and subcutaneous inguinal white adipose tissue (SUB) to body weight were also significantly reduced in a dose-dependent manner, while the ratio of brown adipose tissue (BAT) to body weight was not significantly affected (Figure 1C). Among all groups, the proportions of other organs in body weight did not show significant difference (Figure 1D). Treatment of PO significantly prevented the body weight gain induced by HFD and restrained the accumulation of white adipose tissue (WAT).

### 3.2. PO Maintains Glucose Homeostasis and Reduces Blood Lipid Level

The results of GTT and ITT showed that PO increased glucose tolerance and insulin sensitivity in mice (Figure 2A–D). Compared with HFD group, the areas under the curve (AUC) of GTT and ITT in PO intervention groups were reduced in a dose-dependent manner. The AUC decreased with the increase of dosage and the AUC of ITT in HFD + POM or HFD + POM was significantly lower than that in HFD + POL. In addition, PO also significantly reduced the blood glucose levels in fasting mice (Figure 2E). The blood biochemical results showed that CHO, TG and LDL were markedly lowered by PO (Figure 2F). These results indicates that PO can effectively improve glucose homeostasis and reduce blood lipid while preventing obesity.

### 3.3. PO Has No Adverse Effects in the Liver and Kidney Functions

To explore whether PO had adverse effects on liver and kidney function in mice, the liver/kidney serum biomarkers were measured. The levels of ALT and AST in PO treatment groups and LFD group were significantly lower than those in HFD group and the values in PO treatment groups were close to those in LFD group (Figure 3A), which indicated that PO was likely beneficial for the liver function. Additionally, the levels of ALP, TP and ALB did not show any significant difference (Figure 3B), indicating that PO did not cause any liver damage. Furthermore, UA, BUN and CRE were used to evaluate the function of kidney, and there was no significant difference among five groups (Figure 3C), signifying PO had no influence on kidney function.

### 3.4. PO Beneficially Modulates the Gut Microbiota

It has been demonstrated that there is a close relationship between gut microbiota and obesity, so we explored the effect of PO on the gut microbiota. NMDS and PCA were carried out to compare the microbial community structure. The results revealed a distinct clustering of microbial structure for each group and the gut microbiota compositions of PO treatment groups were more similar to that of LFD group (Figure 4A,B), indicating PO altered the microbiota composition of obese mice fed with HFD towards that of lean mice fed with LFD. The microbial community diversity was analyzed through alpha diversity and beta diversity. Simpson alpha diversity (Figure 4C) and Shannon alpha diversity (Figure 4D) were used to evaluate the gut microbiota richness in samples and the results showed that LFD group had a richer gut microbiota than HFD group, but the treatments of PO did not increase microbiota community diversity obviously. Bray-Curtis beta diversity was used to evaluate the microbiota diversity among samples in every group. The treatments of PO partially decreased the microbiota diversity among samples (Figure 4E). The low and medium dose PO significantly reduced the beta diversity, while high dose PO did not. There was no significant difference among the three different dose groups.

To explore how PO supplementation affected the gut microbiota composition, the microbiota profile at phylum level was analyzed. PO supplementation mainly increased the relative abundance of Bacillota and decreased the relative abundances of Pseudomonadota and Bacteroidota. The gut microbiota profiles of mice in PO treated groups were more similar to that in LFD group (Figure 5A), which indicated that PO can beneficially regulate the gut mocrobiota of obese mice. The correlation between gut microbiota at the phylum level and the body weight of mice was analyzed by the Spearman correlation coefficient. The relative abundances of Bacillota, TM7 and Actinomycetota were significantly negatively correlated with the body weight (Figure 5B,D,F), while the relative abundances of Bacteroidota was significantly positively correlated with the body weight (Figure 5J). Besides, there was a slight negative correlation between the relative abundances of Mycoplasmatota and the body weight (Figure 5H), and there was a slight positive correlation between the relative abundances of Pseudomonadota and the body weight (Figure 5L). Compared with HFD, LFD resulted in more abundant Bacillota, TM7, Actinomycetota and Mycoplasmatota, and the relative abundances of these bacteria were increased by PO intervention in a dose-dependent manner (Figure 5C,E,G,I). In contrast, the relative abundances of Bacteroidota and Pseudomonadota in LFD group were lower than those in HFD group and PO intervention reduced their relative abundances in a dose-dependent manner (Figure 5K,M). In order to explore the differences in specific taxa among different groups, the gut microbiota profile of every mouse was demonstrated in the heat map (Appendix A).

Next, LEfSe analysis was performed to explore the biomarkers of gut microbiota in each group. LEfSe analyses between HFD group and each other group (LFD, HFD + POL, HFD + POM and HFD + POH) were carried out respectively. The taxa from phylum to genus level with significant differences were listed. We analyzed the families and genera whose relative abundances in HFD group were significantly different from those in LFD group or PO treatment groups in particular. By comparison of HFD and LFD, it was found that 20 families and 32 genera were significantly different. Among them, 13 families and 18 genera were significantly higher in LFD group while 7 families and 14 genera were significantly higher in HFD group (Figure 6A,B). Compared with HFD group, HFD + POL significantly altered 18 families (11 increased and 7 decreased) and 30 genera (16 increased and 14 decreased) (Figure 6C,D). As for HFD + POM group, 9 families and 18 genera were increased, while 13 families and 18 genera were reduced (Figure 6E,F). In HFD + POH group, the relative abundances of 25 families and 44 genera were significantly different from those in HFD group. HFD + POH group was significantly richer in 13 families and 19 genera while HFD group was significantly richer in 12 families and 25 genera (Figure 6G,H).

Additionally, the gut bacteria at the family and genus level with significant differences between HFD group and other groups were further analyzed by Venn diagram to find the bacteria with the same trend caused by different dosages of PO and LFD. At the family level, there were five families (Ruminococcaceae, Peptococcaceae, Leuconostocaceae, Eubacteriaceae and Mogibacteriaceae) increased by all of the three PO treatments with different dosages, and four of the five families were also more abundant in LFD group than in HFD group (Figure 7A,C–F). There were also five families decreased in all the three PO treatments groups, including Desulfovibrionaceae, Helicobacteraceae, Beijerinckiaceae, Prevotellaceae and Paraprevotellaceae. Among the five families, three families were also less abundant in the LDF group than in the HFD group (Figure 7B,G–I). It indicated that PO treatment mainly altered the relative abundances of seven families above to mimic those in LFD group. At the genus level, it was found that 10 genera (*Oscillospira*, *Lactobacillus* group, *Bifidobacterium*, *Anaerostipes*, *Anaerovorax*, *Anaerofustis*, *Ruminococcus*, *Coprococcus*, *Bilophila* and *Bacillus*) were all enriched in three PO-treated groups compared with HFD group. Among these genera, six genera were more abundant in LFD group than in HFD group (Figure 8A,C–H). On the other hand, the relative abundances of 11 genera were lowered in all the three PO-treated groups. They were *Bacteroides*, *Roseburia*, *Acinetobacter*, *Agrobacterium*, *Microbacterium*, *Novosphingobium*, *Streptococcus*, *Prevotella*, *Sphingomonas*, *Macrococcus* and *Lactococcus*. Six of the eleven genera were also less abundant in LFD group than in HFD group (Figure 8B,I–N). The results at the genus level revealed that PO supplementation could beneficially modulate the proportions of 12 genera to make their relative abundances closer to those in positive control group, and with the increase of the dosage of PO, the effect was more obvious. Especially for *Oscillospira*, *Lactobacillus*, *Roseburia* and *Acinetobacte*, high-dose PO was more effective than low-dose PO.

### 3.5. PO Regulates the Function of Gut Microbiota

To further explore the relationship between gut microbiota and obesity inhibition, PICRUSt algorithm was used to infer metagenomes based on 16S rRNA gene data and predict the potential function of gut microbiota. PO treatments mainly increased 25 of the Kyoto Encyclopedia of Genes and Genomes (KEGG) pathways. The relative reads count of the 25 pathways was also higher in the LFD group than HFD (Figure 9A). Four of these pathways were involved in lipid metabolism, including synthesis and degradation of ketone bodies, secondary bile acid biosynthesis, primary bile acid biosynthesis and fatty acid metabolism. Three pathways were related to carbohydrate metabolism, including butanoate metabolism, propanoate metabolism, amino sugar and nucleotide sugar metabolism.

On the other hand, there were 24 KEGG pathways that were depressed by PO treatments. These 24 pathways were also lower in LFD group than those in HFD (Figure 9B). Some depressed pathways were interconnected with obesity, such as adipocytokine signaling pathway and steroid hormone biosynthesis. Additionally, some of them were associated with human diseases, such as viral myocarditis, colorectal cancer, small cell lung cancer and Huntington’s disease.

Next, the correlations between body weight, main differential bacteria and KEGG pathways were analyzed by Spearman’s correlation analysis. Body weight was negatively correlated with the increased bacteria and the increased pathways, and the increased bacteria were positively correlated with the increased pathways (Figure 9C). Synthesis and degradation of ketone bodies pathway was significantly positively related to the relative abundances of *Oscillospira*, *Lactobacillus* group, *Bifidobacterium*, *Anaerostipes* and *Anaerovorax*. Primary bile acid biosynthesis and secondary bile acid biosynthesis were positively associated with *Lactobacillus* group, *Bifidobacterium* and *Anaerofustis*. Butanoate metabolism was positively correlated with *Oscillospira*, *Anaerostipes* and *Anaerovorax*. It indicated that these bacteria might promote weight loss. Furthermore, there was a positive correlation between body weight and the decreased bacteria or the decreased pathways. These decreased pathways were positively correlated with the decreased bacteria (Figure 9D). Adipocytokine signaling pathway was significantly positively related to *Bacteroides*, *Acinetobacter*, *Agrobacterium* and *Microbacterium*, and steroid hormone biosynthesis was positively associated with *Roseburia*, *Acinetobacter* and *Agrobacterium*. It meant the restrictions of these bacteria might be beneficial to preventing obesity.

## 4. Discussion

PO, as a common edible mushroom, has many potential beneficial functions. In this research, PO showed an effective anti-obesity activity. The body weight and adipose tissue mass of HFD-induced obese mice were significantly reduced by PO intervention. At the same time, glucose homeostasis was also effectively improved. Considering the close relationship between gut microbiota and obesity and the existing reports of PO regulating gut microbiota, we further explored the effect of PO on gut microbiota in mice and the relationship between the regulation of gut microbiota and the prevention of obesity. The results indicated that PO could modify the gut microbiota in obese mice to approach that in lean mice. To study the relationship between gut microbiota and obesity, the annotation of gut microbiota function was performed. Some functional pathways of gut microbiota related to obesity were regulated by PO, which proved that the effect of PO on obesity is mediated at least partly by the gut microbiota.

In this study, at the phylum level, PO mainly increased Bacillota and TM7, and decreased Pseudomonadota and Bacteroidota. Further analysis showed that PO could beneficially modulate the gut microbiota by enriching six genera. Among the six genera, *Oscillospira*, *Lactobacillus* group and *Bifidobacterium* accounted for a high proportion of gut bacteria which possibly played a more important role than other bacteria. Especially the *Oscillospira* and *Lactobacillus* group, the proportions of *Oscillospira* in all identified intestinal bacteria were 4.43%, 16.14%, 16.95%, 20.77% and 9.02% in HFD, HFD + POL, HFD + POM, HFD + POH and LFD respectively. *Lactobacillus* group accounted for 0.27%, 0.66%, 1.24%, 3.94% and 4.12% in HFD, HFD + POL, HFD + POM, HFD + POH and LFD respectively. As the main abundant intestinal bacteria, *Oscillospira* belongs to Ruminococcaceae, and both *Oscillospira* and *Lactobacillus* group belong to Bacillota, which mainly contributed to the increase of Ruminococcaceae and Bacillota. The phylum Actinomycetota containing *Bifidobacterium* was also enriched by PO supplementation and LFD. It has been documented that *Oscillospira*, *Lactobacillus* group and *Bifidobacterium* are probiotics that can promote weight loss [19,44,45,46,47]. The effect of PO on ameliorating obesity may be mediated by increasing them. On the other hand, PO could also modify the gut microbiota of mice fed with HFD towards normal by reducing six genera. Among the six genera, *Bacteroides* and *Roseburia* had a relatively higher proportion in gut microbiota, so their alterations were more important. Especially *Bacteroides*, the proportion in all identified gut bacteria reached 6.99%, 0.21%, 0.14%, 0.16%, and 0.63% in HFD, HFD + POL, HFD + POM, HFD + POH and LFD respectively. *Bacteroides* is one of the major members of Bacteroidota and accounted for 36.98%, 7.51%, 8.74%, 10.65% and 7.25% in HFD, HFD + POL, HFD + POM, HFD + POH and LFD respectively, so the decrease of Bacteroidota is mainly due to the decrease of *Bacteroides*. In summary, PO intervention modulated the gut microbiota of obese mice induced by HFD and make it similar to the gut microbiota of lean mice fed with LFD. The main bacteria PO altered included 7 families (4 increased and 3 decreased) and 12 genera (6 increased and 6 decreased). The effect of PO on HFD-induced obesity was probably due to the role of PO in beneficially modulating the composition of gut microbiota, typically involving *Oscillospira*, *Lactobacillus* group and *Bifidobacterium*.

The function of PO in regulating the gut microbiota is mediated by its various bioactive components. Like other mushrooms, PO can play the role of prebiotics, which is mainly responsible by its polysaccharides. Previous studies have shown that the important sources of prebiotics in mushrooms are non-digestible mushroom polysaccharides which can inhibit pathogen proliferation by enhancing the growth of probiotic bacteria in the gut [48,49,50]. PO used in this study contains 3.58 ± 0.29% polysaccharides (Appendix A) which might be the main active components modifying the gut microbiota. Of course, further exploration is required before clarifying this view.

To explore whether the function of PO in ameliorating obesity is achieved by regulating the gut microbiota, the functions of gut microbiota were predicted and the correlations between body weight, main differential bacteria and KEGG pathways were analyzed. The synthesis and degradation of ketone bodies and fatty acid metabolism were increased by PO. Ketone bodies are the intermediate metabolites in fat oxidation and metabolism. The increase of synthesis and degradation of ketone bodies and fatty acid metabolism revealed the enhancement of fat motivation. The biosynthesis of primary bile acid and secondary bile acid was both increased in the PO-treated groups. Bile acids can promote energy expenditure through the TGR5–cAMP–D2 signal pathway [27], mitochondrial fission and beige remodeling of white adipose tissue through ERK/DRP1 pathway [28]. The butanoate metabolism and propanoate metabolism were also increased by PO. Both butanoate and propanoate have shown to prevent obesity [51,52]. On the other hand, PO reduced the adipocytokine signaling pathway, steroid hormone biosynthesis and some pathways related to human diseases. There is a close link between adipocytokines and obesity or metabolic diseases. Many adipocytokines, including TNF-α, Resistin and IL-6, positively correlate with obesity [53,54]. Steroid hormones are involved in the metabolism, accumulation and distribution of adipose tissues and are positively correlated with obesity [55]. Therefore, the reduction of adipocytokine signaling and steroid hormone biosynthesis contributed to weight loss. Some of the depressed pathways were associated with human diseases (viral myocarditis, colorectal cancer, small cell lung cancer, Huntington’s disease), whose decreases revealed that PO had a beneficial effect on health. In summary, PO increased lipid metabolism, carbohydrate metabolism, bile acid biosynthesis and decreased adipocytokine signaling pathway and steroid hormone biosynthesis. Correlation analysis suggested *Oscillospira*, *Lactobacillus* group and *Bifidobacterium* might play an important role in increasing lipid metabolism, carbohydrate metabolism, bile acid biosynthesis, and *Bacteroides* and *Roseburia* might play an important role in decreasing adipocytokine signaling pathway and steroid hormone biosynthesis. These findings were further confirmed by some existing studies. For example, *Oscillospira* has been reported to produce SCFAs and prevent the fat accumulation [46]. *Lactobacillus* group has been reported to be involved in carbohydrate transport and metabolism and can ameliorate obesity [56,57], and *Bifidobacterium* has been shown to modulate lipid metabolism [58]. So, *Oscillospira*, *Lactobacillus* group and *Bifidobacterium* may be mainly responsible for the role of PO in ameliorating obesity, but this still needs to be further verified.

In summary, this study proved that PO could prevent obesity and beneficially modulate gut microbiota. All three dosages of PO had significant effects, and the effect was more significant with the increase of PO dosage. The composition and functions of gut microbiota in mice fed with HFD were altered towards those fed with LFD by PO, and the alterations of the composition and functions of gut microbiota caused by PO contributed to weight loss. Therefore, PO ameliorated obesity at least partly by modulating gut microbiota.

## Figures and Tables

**Figure 1 nutrients-14-01868-f001:**
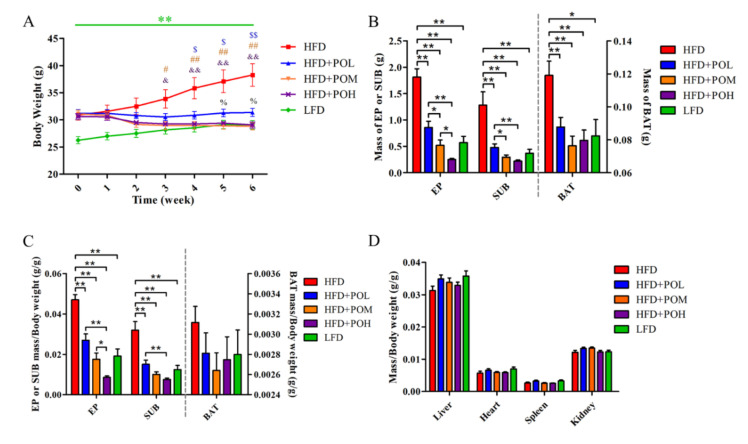
PO prevents HFD-induced obesity in mice. (**A**) Body weight. * comparison between HFD and LFD, $ comparison between HFD and HFD + POL, # comparison between HFD and HFD + POM, & comparison between HFD and HFD + POH, % comparison between HFD + POL and HFD + POM or HFD + POH. (**B**) Mass of adipose tissue. (**C**) Ratio of adipose tissue to body weight. (**D**) Ratio of organs to body weight. Data are shown as mean ± SEM (*n* = 8). *, $, #, &, % *p* < 0.05; **, $$, ##, && *p* < 0.01.

**Figure 2 nutrients-14-01868-f002:**
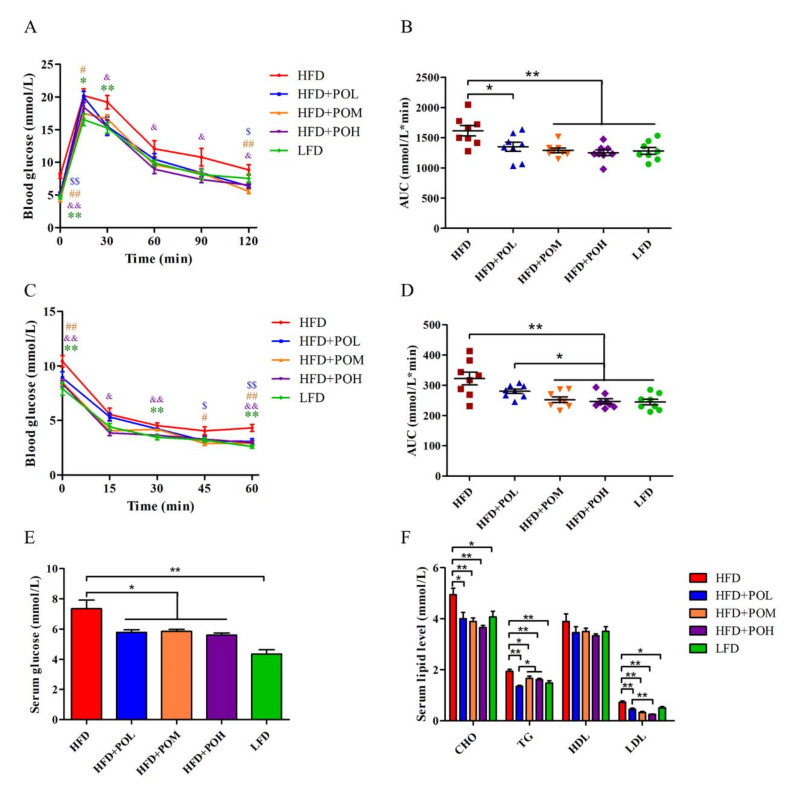
PO maintains glucose homeostasis and reduces blood lipid level. (**A**) GTT. (**B**) Area under curve of GTT. (**C**) ITT. (**D**) Area under curve of ITT. (**E**) Fasting blood glucose. (**F**) Blood lipid. (**A**,**C**) * comparison between HFD and LFD, $ comparison between HFD and HFD + POL, # comparison between HFD and HFD + POM, & comparison between HFD and HFD + POH. Data are shown as mean ± SEM (*n* = 8). *, $, #, & *p* < 0.05; **, $$, ##, && *p* < 0.01.

**Figure 3 nutrients-14-01868-f003:**
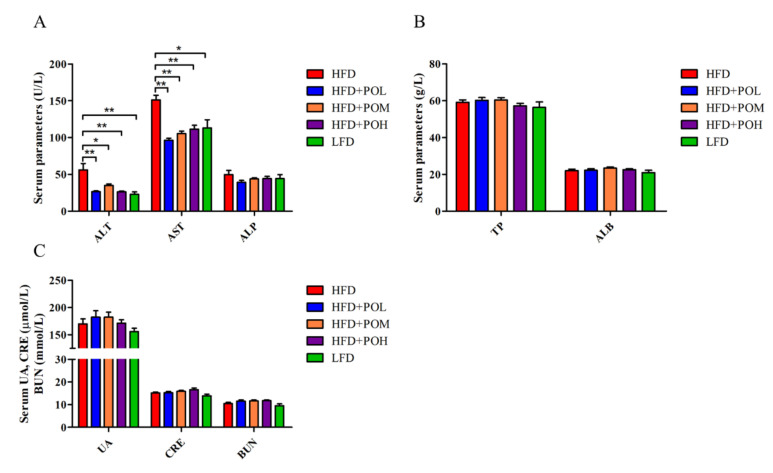
PO has no obvious harm to the liver and kidney. (**A**) Liver function biomarker (ALT, AST, ALP). (**B**) Liver function biomarker (TP, ALB). (**C**) Kidney function biomarker. Data are shown as mean ± SEM (*n* = 8). * *p* < 0.05, ** *p* < 0.01 compared with HFD.

**Figure 4 nutrients-14-01868-f004:**
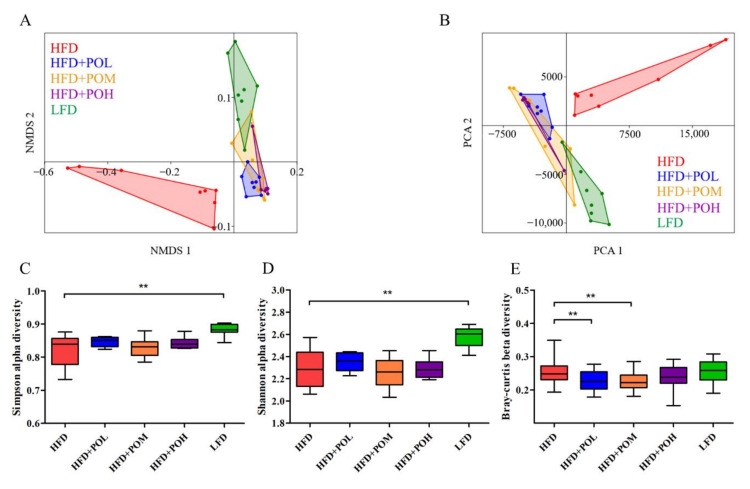
NMDS analysis, PCA analysis and diversity analysis. (**A**) NMDS analysis; (**B**) PCA analysis; (**C**) Simpson alpha diversity 1-D; (**D**) Shannon alpha diversity; (**E**) Bray-Curtis beta diversity. Data are shown as mean ± SEM (*n* = 6–8). ** *p* < 0.01 compared with HFD.

**Figure 5 nutrients-14-01868-f005:**
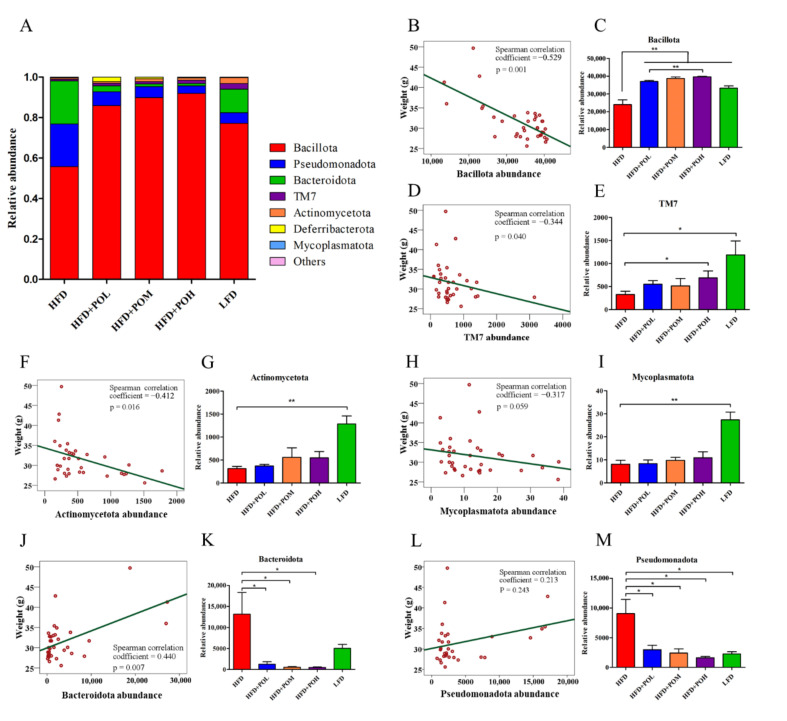
PO beneficially modulates the composition of gut microbiota. (**A**) Microbiota profile at phylum level; (**B**–**M**) Correlation analysis between the relative abundance of gut microbiota at phylum level and body weight. Data are shown as mean ± SEM (*n* = 6–8). * *p* < 0.05, ** *p* < 0.01.

**Figure 6 nutrients-14-01868-f006:**
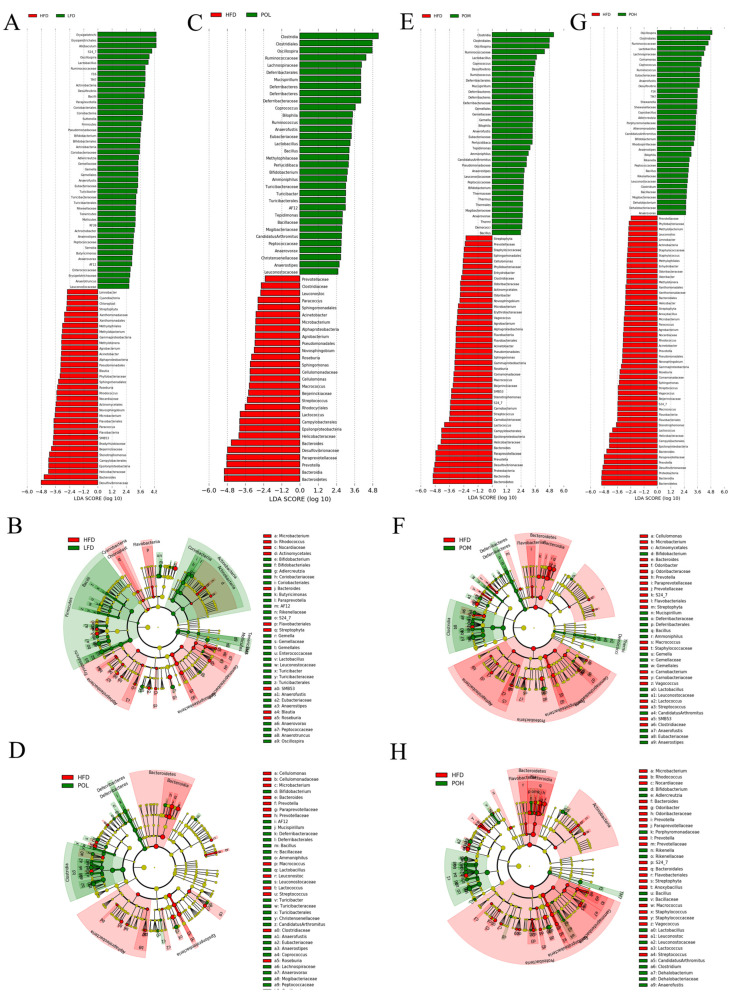
LEfSe analysis. (**A**,**B**) Biomarker taxa and cladogram between HFD and LFD; (**C**,**D**) Biomarker taxa and cladogram between HFD and HFD + POL; (**E**,**F**) Biomarker taxa and cladogram between HFD and HFD + POM; (**G**,**H**) Biomarker taxa and cladogram between HFD and HFD + POH. *n* = 6–8.

**Figure 7 nutrients-14-01868-f007:**
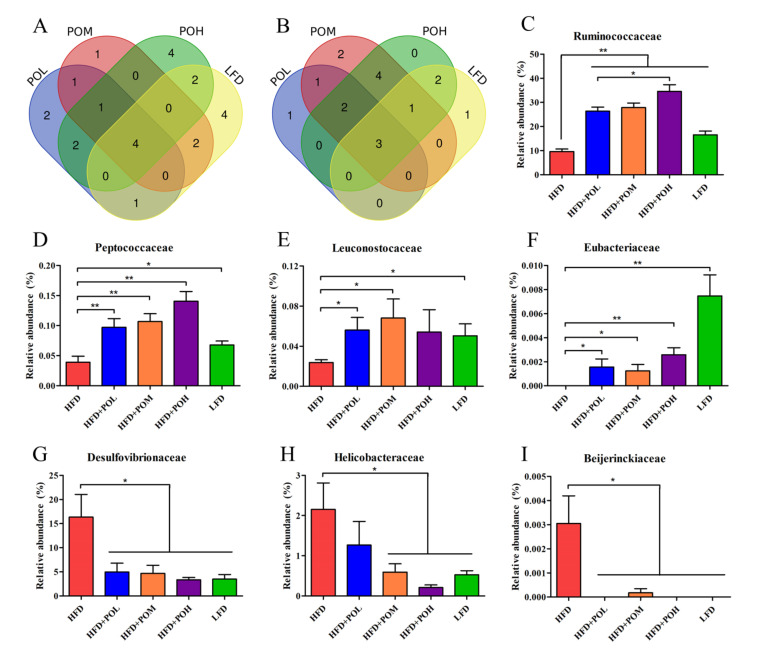
Gut bacteria at the family level showed significant differences between HFD group and other groups. (**A**) Venn diagram of increased families in PO treatment groups and LFD compared with HFD; (**B**) Venn diagram of decreased families in PO treatment groups and LFD compared with HFD; (**C**–**F**) The relative abundance of families increased by all of the three PO treatments and LFD; (**G**–**I**) The relative abundance of families decreased by all of the three PO treatments and LFD. Data are shown as mean ± SEM (*n* = 6–8). * *p* < 0.05, ** *p* < 0.01.

**Figure 8 nutrients-14-01868-f008:**
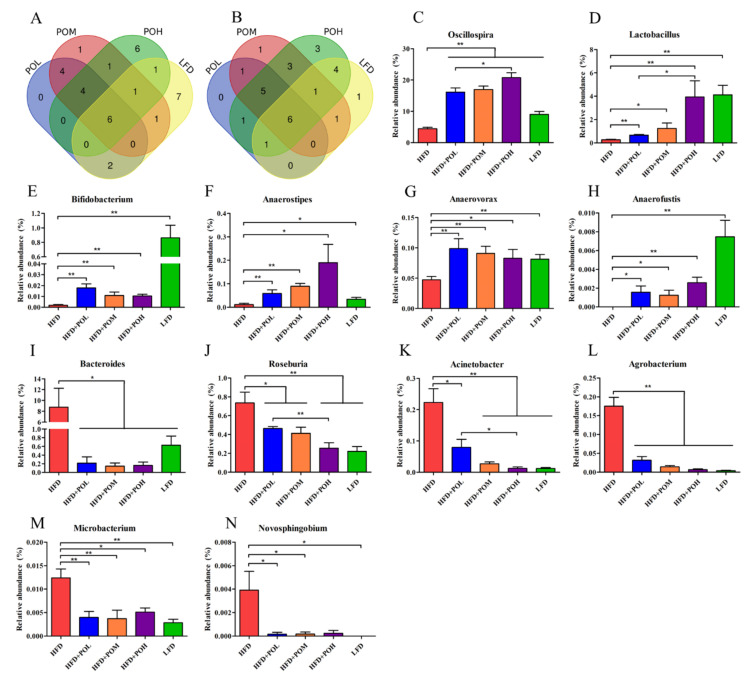
Gut bacteria at the genus level showed significant differences between HFD group and other groups. (**A**) Venn diagram of increased genera in PO treatment groups and LFD compared with HFD; (**B**) Venn diagram of decreased genera in PO treatment groups and LFD compared with HFD; (**C**–**H**) The relative abundance of genera increased by all of the three PO treatments and LFD; (**I**–**N**) The relative abundance of genera decreased by all of the three PO treatments and LFD. Data are shown as mean ± SEM (*n* = 6–8). * *p* < 0.05, ** *p* < 0.01.

**Figure 9 nutrients-14-01868-f009:**
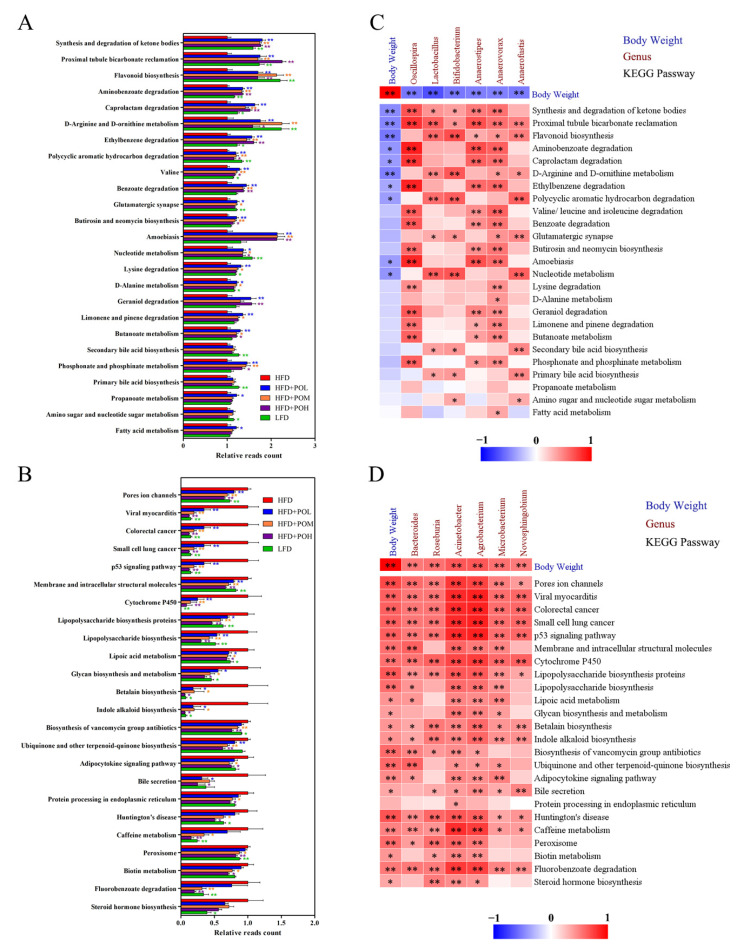
Prediction of gut microbiota function. (**A**) KEGG pathways increased by PO; (**B**) KEGG pathways decreased by PO; (**C**) Spearman’s correlation among body weight, main increased bacteria and increased KEGG pathways; (**D**) Spearman’s correlation among body weight, main decreased bacteria and decreased KEGG pathways. Data are shown as mean ± SEM (*n* = 6–8). * *p* < 0.05, ** *p* < 0.01.

## Data Availability

Data are available from the corresponding author on reasonable request.

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
