# Peer review of "Pleurotus Ostreatus Ameliorates Obesity by Modulating the Gut Microbiota in Obese Mice Induced by High-Fat Diet"

_nutrients, 2022, doi:10.3390/nu14091868_

Round 1

Reviewer 1 Report

Congratulations to the authors of a very interesting work. My suggestions are in the attached file in the form of comments and highlights.

Reviewer 2 Report

The aim of the research article by Hu et al is to evaluate the potential anti-obesity properties of the administration of different doses of the mushroom Pleurotus ostreatus (PO) in an in vivo model of diet induced obesity. The authors have also tested the modulation of the gut microbiota by the administration of this edible mushroom. The article is in general easy to follow but it would be desirable to be revised by a native English Speaker because the language and style should be improved. The methods employed are appropriate and the objective is clear but some concerns need to be addressed.  it is necessary to justify the different doses of PO employed, to better explainand describe the gut microbiota data and  conclude if it is better to administrate a specific dose of PO for ameliorate obesity.

General comments

-I would cite in the introduction section if another edible mushrooms close to PO have been tested employing in vivo models for their potential antiobesity properties to justify the objective and the novelty of the work

-It would be useful that the authors explain and justify why three different doses of PO mass fractions were tested

-Please take care about the new nomenclature of Prokaryote Phyla. For example, Bacteroidetes Phylum is now called Bacteroidota. On December 2021 the National Centre for Biotechnology Information (NCBI), which specifies the standard names of all organisms in other public databases, adopted a new system to name the phylum of some prokaryotes. Please check and change in the text and figures accordingly.

-The authors in the different analyses performed they always compared with HFD group but if different doses of PO have been tested they should also mention and better explain differences between the groups HFD-PL. HFD-POM and HFD-POH. Checking the figures, I would expect differences between PO treated groups and these data are important and have not been considered.

Specific comments

Material and methods section:

-Please indicate how the composition of the PO power was measured

-Regarding the ITT in which week please specify when it was performed

Results section:

-In Figure 1A is depicted different symbols but in the legend from the figure is only indicated * P<0.05 and ** p<0.01 compared with group HF, please better explain the other symbols employed in this figure.

-Figure 1B I would not include and just say the results in 31.1 section or in case the authors want to leave this figure I would include as supplementary information. Please do not include ns because they should also use the same nomenclature in figure 1E and other figures were not significant changes in the parameters tested were observed.

-Please change the title 3.3 for PO has no adverse effects in the liver and kidney functions.

Section 3.4: This section is too long…I would describe the main results and in a more precise way. In addition some sentences in this section it would be better to be included in discussion section. For example PO could beneficially modulate the gut microbiota by enrichening 6 genera…and it has been documented that Oscillospira, Lactobacillus and Bifidobacterium are probiotics…Please check and re-write this section and shorten it.

Section 3.5:  Some sentences are not results and they should better be included in discussion section. For example many adypokines are positively associated with obesity (46-47) or steroid hormones are also positively associated with obesity (48)…

-Moreover indicate the PO groups that significantly decreased the beta diversity and as I have previously mentioned It would be useful to have information about differences between PO treated groups in this section at phylum and family taxonomic level.

-Regarding the gut microbiota at phyla ltaxonomic the authors indicate that PO supplementation mainly increased the relative abundance of Firmicutes and decreased the relative abundance of Proteobacteria and Bacteroidetes whereas the microbiota profiles in the PO treated mice were more similar to LFD group but it is not clear if these results are significative or they are only a description of the profile at this taxonomic level. Please better explain.

In Figures 5B, D, H the authors have depicted correlation analyses but in the case of 5H with Tenericutes they cannot say that there was a significative negative correlation with body weight because p value is 0.059. In figure 5J with Proteobacteria is the same, it is not a significative positive correlation. Please clearly indicate in this results section when the data are significative to be rigorous.

-Figure 5N: I would include in supplementary information.

Discussion section:

I would avoid to use the ratio Firmicutes/ Bacteroidetes because as the authors have mentioned is controversial

Regarding Lactobacillus I would use Lactobacillus group because new nomenclature of this group has been recently reviewed (please see https://doi.org/10.1099/ijsem.0.004107)

Please do not say that Akkemansia muciniphila belongs to Firmicutes because it is not correct and belongs to Verrucomicrobia phylum and check the sentences related to the increase of Firmicutes phylum.
